# AutoM³L: An Automated Multimodal Machine Learning Framework with Large Language Models

## ABSTRACT

Automated Machine Learning (AutoML) offers a promising approach to streamline the training of machine learning models. However, existing AutoML frameworks are often limited to unimodal scenarios and require extensive manual configuration. Recent advancements in Large Language Models (LLMs) have showcased their exceptional abilities in reasoning, interaction, and code generation, presenting an opportunity to develop a more automated and user-friendly framework. To this end, we introduce AutoM³L, an innovative Automated Multimodal Machine Learning framework that leverages LLMs as controllers to automatically construct multimodal training pipelines. AutoM³L comprehends data modalities and selects appropriate models based on user requirements, providing automation and interactivity. By eliminating the need for manual feature engineering and hyperparameter optimization, our framework simplifies user engagement and enables customization through directives, addressing the limitations of previous rule-based AutoML approaches. We evaluate the performance of AutoM³L on six diverse multimodal datasets spanning classification, regression, and retrieval tasks, as well as a comprehensive set of unimodal datasets. The results demonstrate that AutoM³L achieves competitive or superior performance compared to traditional rule-based AutoML methods. Furthermore, a user study highlights the user-friendliness and usability of our framework, compared to the rule-based AutoML methods. Code is available at: https://anonymous.4open.science/r/anonymization_code.

## CCS CONCEPTS

• **Computing methodologies → Artificial intelligence**; **Machine learning**; • **Human-centered computing → Human computer interaction (HCI)**.

## KEYWORDS

human-AI interaction, automated machine learning, large language model, usability, user study

## 1 INTRODUCTION

Multimodal data is crucial in various machine learning (ML) tasks as it provides the ability to capture more comprehensive feature representations. Real-world data often combines heterogeneous sources, such as integrating table product information with associated images and textual descriptions. Similarly, in the financial sector, user photos, text, transactions, and other data types are frequently consolidated in tabular formats for analysis and management. However, the inherent diversity of these modalities introduces complexities, particularly in selecting optimal machine learning or deep learning model architectures and seamlessly synchronizing features across modalities Consequently, there is often a heavy reliance on manual involvement in the ML pipeline.

Automated Machine Learning (AutoML) has emerged as a promising approach to reduce the need for manual intervention in the ML pipeline [6, 10, 16, 30, 31, 33]. However, a significant gap exists for multimodal data, as the majority of AutoML solutions primarily focus on unimodal data. AutoGluon[1] made an initial attempt at multimodal AutoML but suffers from several limitations. Firstly, it lacks comprehensive automation of feature engineering, which is crucial for effectively handling multimodal data. Secondly, it presents a steep learning curve for users to become familiar with its configurations and settings, contradicting the user-friendly automation principles that AutoML aims to embody. Moreover, AutoGluon's adaptability is constrained by pre-set settings such as the search space, model selection, and hyperparameters, necessitating significant manual intervention. Lastly, extending AutoGluon's capabilities by integrating new techniques or models often requires complex manual code modifications, hindering its agility and potential for growth.

The scientific community has been captivated by the rapid rise of large language models (LLMs), particularly due to their transformative potential in task automation [2, 4, 29, 34]. LLMs have evolved beyond their initial purpose as text generators and have now become highly autonomous entities capable of self-initiated planning and execution [14, 27, 32, 35, 36]. This evolution presents a compelling opportunity to enhance the performance and adaptability of multimodal AutoML systems. Leveraging this potential, we introduce AutoM³L, an innovative LLM framework for Automated Multimodal Machine Learning. Unlike platforms such as AutoGluon, which are constrained by predefined pipelines, AutoM³L distinguishes itself through its dynamic user interactivity. Specifically, it seamlessly integrates ML pipelines tailored to user instructions, enabling unparalleled scalability and adaptability throughout the entire process, from data pre-processing to model selection and optimization.

The major contributions are four-fold, summarized as follows. (1) We introduce AutoM³L, a novel framework that automates the development of machine learning pipelines for multimodal data. AutoM³L enables users to derive accurate models for each modality from a diverse pool of models and generates an executable script for cross-modality feature fusion, all with minimal natural language instructions. This approach simplifies the process of building multimodal ML pipelines and makes it more accessible to a wider range of users. (2) We advance the automation of feature engineering by leveraging a LLM to intelligently filter out attributes that could hinder model performance while simultaneously imputing missing data. This automated feature engineering process reduces the need for manual intervention and improves the overall quality of the input data. (3) We automate hyperparameter optimization by combining the LLM's self-generated suggestions with external API calls. This approach eliminates the need for labor-intensive manual explorations and enables more efficient and effective hyperparameter tuning. (4) We conduct comprehensive evaluations, comparing AutoM³L with conventional rule-based AutoML on a diverse set of

---

[1] https://github.com/autogluon/autogluon

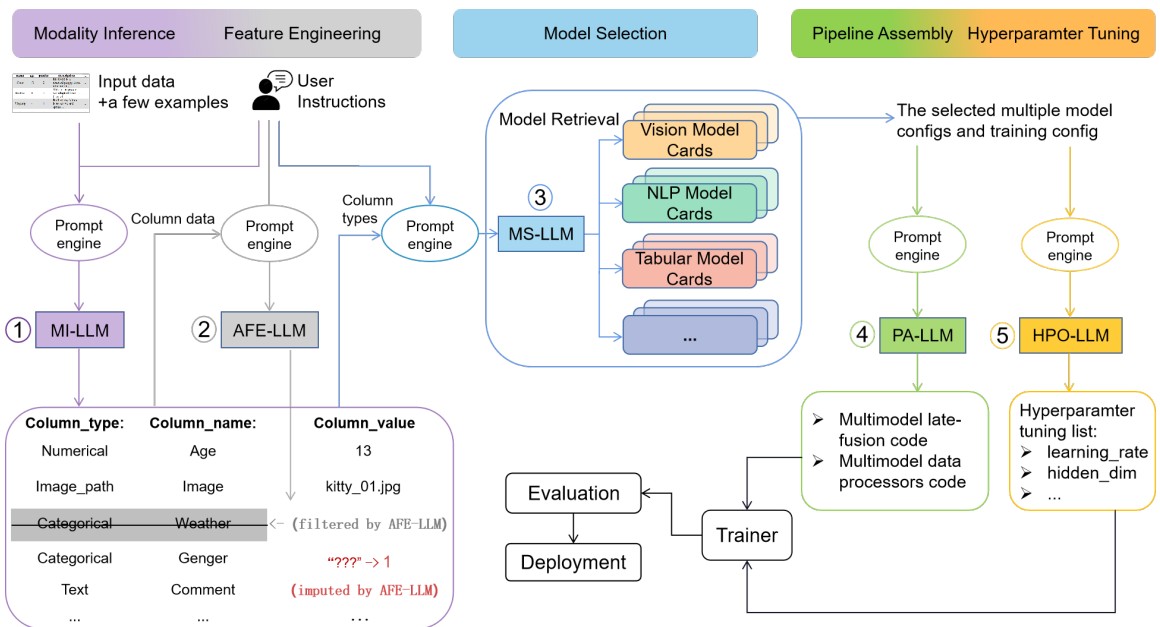

Figure 1: The overall framework of AutoM³L. It consists of five stages: ① Infer the modality of each attribute in structured table data. ② Automate feature engineering for feature filtering and data imputation. ③ Select optimal models for each modality. ④ Generates executable scripts for model fusion and data processing to assemble the training pipeline. ⑤ Search optimal hyperparameters. The detailed system prompts for LLMs in each stage can be found in Appendix A.

multimodal and unimodal datasets. Additionally, a user study further highlighted the distinct advantages of our framework in terms of user-friendliness and a significantly reduced learning curve.

## 2 METHODS

In this paper, we propose an **Auto**mated **M**ulti-**M**odal **M**achine **L**earning (AutoM³L) framework that utilizes Large Language Models (LLMs) to automate the machine learning pipeline for multimodal scenarios. This section begins by introducing the organization of the multimodal dataset in Sec.2.1. In Sec.2.2 to 2.6, we elaborate on the five functional components enhanced by LLMs in AutoM³L: (1) modality inference, (2) automated feature engineering, (3) model selection, (4) pipeline assembly, and (5) hyperparameter optimization, as illustrated in Fig. 1.

### 2.1 Organization of Multimodal Dataset

Most existing studies utilize the JavaScript Object Notation (JSON) to represent multimodal data. However, JSON cannot capture the interplay between different modalities, making it unsuitable for analysis by language models. To address this limitation, we follow [3, 12, 15, 28] and employ the structured tables to represent multimodal data. Structured tables offer a clear representation that captures the interaction between different modalities and effectively aggregates information from various formats into a unified structure. Additionally, these tables encompass a diverse range of data modalities, including images, text, tabular data, and more.

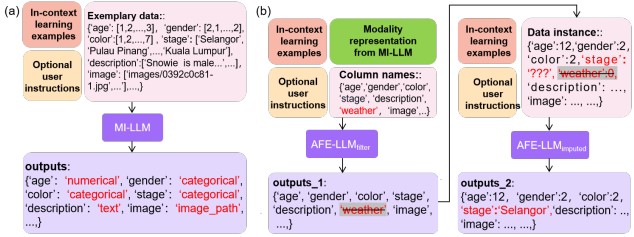

Figure 2: (a) Modality Inference with MI-LLM. It displays MI-LLM's capability to determine the modality of each column in a dataset. Attributes are annotated in red to indicate the inferred modality. (b) Data Refinement with AFE-LLM. It highlights AFE-LLM's dual role in feature filtering and data imputation. The left part displays attributes marked in red that are filtered out, while the right part shows red annotations identifying attributes that undergo imputation.

### 2.2 Modality Inference Module

AutoM³L begins with the **M**odality **I**nference-LLM (MI-LLM) component, which identifies the associated modality for each column in the structured table. To simplify its operation and minimize additional training costs, MI-LLM leverages in-context learning. As illustrated in Fig. 2(a), the guiding prompt for MI-LLM consists of three essential parts: (1) An ensemble of curated examples is utilized for in-context learning, assisting MI-LLM in establishing strong correlations between column names and their associated modalities, thereby generating the desired format responses. These examples

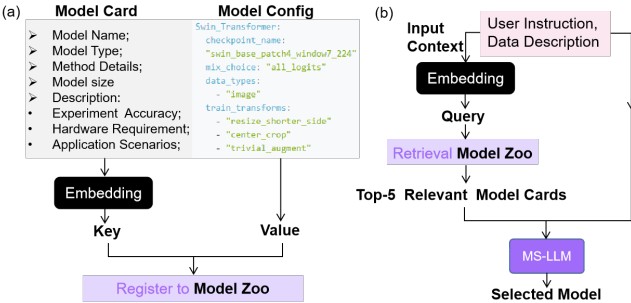

**Figure 3: Illustration of the model zoo and MS-LLM. (a) Model addition process: This stage showcases how new models are incorporated into the model zoo, visualized as a vector database. The model card's embedding vector serves as the unique identifier or key, paired with its corresponding model configuration as the value. (b) Model retrieval process: This stage illustrates the model selection process. Given user directives, the system initiates a query to identify the top 5 models that align with each input modality. From this refined subset, MS-LLM then determines and selects the most suitable model.**

serve as a foundation for MI-LLM to learn from and adapt to the specific dataset at hand, enabling it to accurately infer the modality of each column based on the provided examples. (2) A subset of the input structured table, consisting of randomly sampled data items paired with their respective column names, is included. The semantic richness of this subset acts as a guiding force, directing MI-LLM towards accurate identification of modalities. By providing a representative sample of the dataset, MI-LLM can better understand the context and characteristics of each column, allowing it to make more informed modality inferences. (3) User directives go beyond mere instructions, enriching the process with deeper contextual information. These directives leverage the LLM's exceptional interactivity to enhance the refinement of modality inference. For instance, a directive such as "*This dataset delves into the diverse factors influencing animal adoption rates*" provides MI-LLM with valuable contextual information, enabling a more insightful interpretation of column descriptors. This additional context helps MI-LLM to make more accurate and relevant modality inferences by considering the overall theme and purpose of the dataset.

### 2.3 Automated Feature Engineering Module

Feature engineering is a critical pre-processing phase to address common data challenges, such as handling missing values. While conventional AutoML solutions rely heavily on rule-based feature engineering, our AutoM³L framework leverages the exceptional capabilities of LLMs to enhance this process. Specifically, we introduce the **A**utomatic **F**eature **E**ngineering-LLM (AFE-LLM), as illustrated in Fig. 2(b). This module utilizes two distinct prompts, resulting in two core components: AFE-LLM$_{filter}$ and AFE-LLM$_{imputed}$. The AFE-LLM$_{filter}$ component effectively sifts through the data to eliminate irrelevant or redundant attributes. On the other hand, the AFE-LLM$_{imputed}$ component is dedicated to data imputation, ensuring the completeness and reliability of essential data. Importantly, these components work together in synergy. After AFE-LLM$_{filter}$

refines the features, AFE-LLM$_{imputed}$ then addresses relevant data gaps in the dataset.

To enhance feature filtering, AFE-LLM$_{filter}$ incorporates the following prompts: (1) An ensemble of examples for in-context learning, which includes introducing attributes from diverse datasets and intentionally incorporating irrelevant attributes. The objective of AFE-LLM$_{filter}$ is to effectively distinguish and eliminate irrelevant attributes (2) Column names in the structured table, containing abundant semantic information about each feature component, thereby enhancing the LLM's capability to distinguish between crucial and dispensable attributes. (3) Modality inference results derived from MI-LLM, guiding the LLM to remove attributes of limited informational significance. For instance, when comparing a binary attribute that indicates whether someone is over 50 with a continuous attribute such as age, it becomes apparent that the binary attribute may be somewhat redundant. In this case, the binary attribute can be identified and removed. (4) User instructions or task descriptions can be embedded when available, aiming to establish a connection between column names and the corresponding task.

On the other hand, the AFE-LLM$_{imputed}$ component is dedicated to data imputation, ensuring the completeness and reliability of essential data. Regarding data imputation, AFE-LLM$_{imputed}$ exploits its inferential capabilities to effectively identify and fill missing data. The prompt for this aspect includes the following: (1) Data points with missing values, enabling AFE-LLM$_{imputed}$ to fill these gaps by discerning patterns and inter-attribute relationships. (2) A selected subset of data instances from the training set that involves deliberately masking individual attributes and presenting them in Q&A pairs, laying down an inferential groundwork. (3) Where available, user instructions or task descriptions are incorporated, offering a richer context and further refining the data imputation process.

Importantly, these components work together in synergy. After AFE-LLM$_{filter}$ refines the features, AFE-LLM$_{imputed}$ then addresses relevant data gaps in the dataset. By combining feature filtering and data imputation, this module ensures that the dataset is optimized for the subsequent steps in the AutoM³L pipeline.

### 2.4 Model Selection Module

Upon successfully performing the modality inference and automated feature engineering modules, AutoM³L proceeds to determine the optimal model architecture for each data modality. The candidate models are cataloged within a model zoo, with each model stored as a model card. The model card captures a wide range of details, including the model's name, type, applicable data modalities, empirical performance metrics, hardware requirements, and other relevant information. To streamline the generation of these cards, we leverage LLM-enhanced tools, such as ChatPaper[37], to eliminate the need for laborious manual writing processes. We generate embeddings for these model cards using a text encoder, thereby allowing users to retrieve relevant model cards and seamlessly expand the model zoo by appending new cards, as illustrated in Fig. 3(a).

Following the model card generation, we propose the **M**odel **S**election-LLM (MS-LLM) to effectively match each modality with the appropriate model. We view this task as a single-choice dilemma,

where the context provides a range of models for selection. However, due to limitations on the context length of LLM, it is not feasible to present a complete array of model cards. Hence, we initially filter the model cards based on their applicable modality type and keep only those that are aligned with the specified data modality. Next, a subset of the top 5 models is identified using text-based similarity metrics to compare the user's requirements with the model cards' descriptions. These high-ranking model cards are then incorporated into the prompt of MS-LLM, along with user instructions and data descriptions. This combination guides MS-LLM in making its final decision, ultimately identifying the most suitable model for the given modality, as illustrated in Fig. 3(b).

The MS-LLM prompt fuses the following components: (1) A selected subset of five model cards, providing insight into potential model candidates. (2) An input context that intertwines data descriptions and user instructions. The data descriptions clarify important aspects such as data type, label type, and evaluation standards. Meanwhile, user instructions can provide clarification on specific model requirements. For example, a user instruction such as "deploy the model on the CPU device" would guide MS-LLM to models optimized for lightweight deployments. This enhances the user-friendliness and intelligence of the framework by enabling interactive execution.

## 2.5 Pipeline Assembly Module

After retrieving the unimodal models, a crucial step involves fusing them. We employ a late fusion strategy for integration, which can be mathematically expressed as:

$$F_i = \texttt{feature\_adapter}_i(\texttt{model}_i(x_i)), \tag{1}$$

$$F_{cat} = \texttt{concat}(F_1, ..., F_n), \tag{2}$$

$$\texttt{logits}_{\texttt{fuse}} = \texttt{fusion\_head}(\texttt{fusion\_model}(F_{cat})), \tag{3}$$

where `concat` denotes concatenation, $x_i$ represents the input data of modality $i$ ($i = 1, \cdots, n$), and `feature_adapter`$_i$ adapts the output of `model`$_i$ to a consistent dimension. The `fusion_head` and `fusion_model` are the target models that need to be built. Determining the architectures for `fusion_head` and `fusion_model` using rule-based methods that require manual scripting is impractical, as the architectures depend on the number of input modalities. Instead, we reframe this process as a code generation challenge, where the **P**ipeline **A**ssembly-LLM (PA-LLM) is responsible for generating the fusion model architecture. PA-LLM leverages the code generation capabilities of LLMs to produce executable code for both model fusion and data processors, as depicted in Fig. 4(a). This is achieved by providing the module with relevant model configuration files within the prompt. The data processors are generated based on the specified data preprocessing parameters in the configuration file. We prioritize the integration of pre-trained models from various modalities, sourced from well-known libraries such as `HuggingFace` and `Timm`. By establishing ties with the wider ML community, we have significantly enhanced the versatility and applicability of our model zoo.

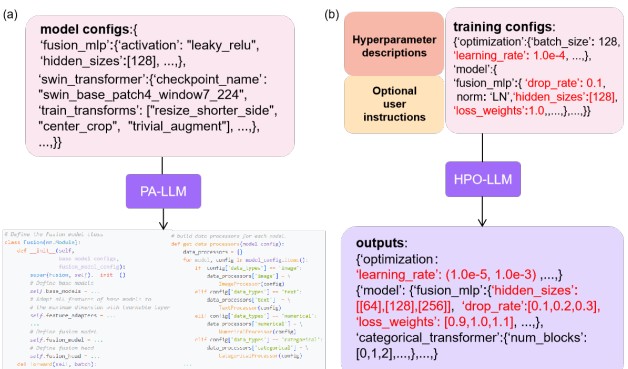

Figure 4: (a) The PA-LLM is responsible for generating executable code, ensuring seamless model training and data processing. (b) On the other hand, the HPO-LLM deduces optimal hyperparameters and defines appropriate search intervals for hyperparameter optimization.

## 2.6 Automated Hyperparameter Optimization Module

In conventional ML pipelines, hyperparameters such as learning rate, batch size, hidden layer size, and loss weight are commonly adjusted manually, which is labor-intensive and time-consuming. Although external tools like `ray.tune` allow users to conduct optimization by specifying hyperparameters and their search intervals, there is still room for further automation. To bridge this gap, we propose the **H**yper**P**arameter **O**ptimization-LLM (HPO-LLM), which extends the foundational capabilities of `ray.tune`. The core functionality of HPO-LLM lies in its ability to determine optimal hyperparameters and their corresponding search intervals through careful analysis of a provided training configuration file, as depicted in Fig. 4(b). Leveraging the extensive knowledge base of LLMs in ML training, we first utilize LLM to generate comprehensive descriptions for each hyperparameter specified in the configuration file. The descriptions, combined with the original configuration file, constitute the prompt for HPO-LLM, which then provides recommendations on the most suitable hyperparameters for optimization. The input prompt provided to HPO-LLM encompasses the following components: (1) The training configuration file, containing a comprehensive set of hyperparameters, assists HPO-LLM in selecting the most suitable hyperparameters for optimization. (2) LLM-generated text descriptions for each hyperparameter, enabling HPO-LLM to gain a comprehensive understanding of the significance of each hyperparameter. (3) Optional user directives provide a personalized touch, allowing users to incorporate additional instructions that guide HPO-LLM's decision-making process. These directives can include emphasizing specific hyperparameters based on unique requirements, resulting in a tailored optimization approach. By integrating the capabilities of `ray.tune` with our HPO-LLM, we have pioneered an approach that enhances hyperparameter optimization by combining automation with advanced decision-making.

**Table 1: Task and structure of multimodal datasets**

| Dataset Name | #Train | #Test | Task | Metric | Prediction Target |
|---|---|---|---|---|---|
| PAP | 13493 | 1499 | multiclass | accuracy | category of adoption speed |
| MMSD | 17833 | 1981 | binary | auc | whether utterances contains an ironic sentiment |
| PPC | 8920 | 993 | regression | rmse | pawpularity score |
| PARA | 25398 | 2822 | regression | rmse | image aesthetics assessment |
| SPMG | 5000 | 1000 | retrieval | auc | whether data pair is in the same class |
| CH-SIMS | 2052 | 228 | multiclass | accuracy | category of sentiment |

**Table 2: Evaluation for modality inference. AutoM³L can effectively determine the data modality, even on data that AutoGluon misclassifies or unclassifies. \* means the result of manual corrections in modality inference.**

| Dataset | AutoGluon | AutoM³L |
|---|---|---|
| PAP↑ | 0.415(0.011) | 0.409(0.014) |
| MMSD↑ | 0.958(0.004) | 0.956(0.004) |
| PPC↓ | 17.78(0.307) | 17.71(0.315) |
| PARA↓ | 0.568(0.019) | 0.571(0.021) |
| SPMG↑ | 0.985(0.003) | 0.986(0.003) |
| CH-SIMS↑ | 0.543(0.032)* | 0.575(0.029) |

**Table 3: Evaluation for feature engineering. AutoM³L filters out noisy features and performs data imputation, effectively mitigating the adverse effects of noisy data. \* means the result of manual corrections in modality inference.**

| Dataset | AutoKeras | AutoGluon | AutoM³L |
|---|---|---|---|
| PAP↑ | 0.379(0.018) | 0.402(0.014) | 0.407(0.012) |
| MMSD↑ | 0.920(0.008) | 0.951(0.004) | 0.956(0.004) |
| PPC↓ | 25.18(0.302) | 18.38(0.298) | 17.82(0.304) |
| PARA↓ | 0.782(0.025) | 0.576(0.020) | 0.574(0.020) |
| SPMG↑ | / | 0.984(0.003) | 0.986(0.003) |
| CH-SIMS↑ | / | 0.540(0.031)* | 0.575(0.029) |

## 3 EXPERIMENTS

### 3.1 Experimental Settings

*3.1.1 Datasets.* To assess the effectiveness of the AutoM³L system, we performed experiments on six multimodal datasets, including some obtained from the Kaggle competition platform. These datasets cover various tasks, such as classification, regression, and retrieval. Table 1 describes the details of the datasets. We utilized three classification datasets as follows: (1) PetFinder.my-Adoption Prediction (PAP): This dataset aims to predict the adoptability of pets by analyzing image, text, and tabular modalities. (2) Multi-Modal Sarcasm Detection (MMSD): This dataset is curated to determine whether an utterance contains ironic sentiment, utilizing image and text modalities. (3) CH-SIMS: This dataset focuses on sentiment recognition and leverages video and text modalities. Turning our attention to regression, we utilized two datasets: (1) PetFinder.my-Pawpularity Contest dataset (PPC): This dataset aims to predict the popularity of shelter pets by leveraging image and tabular modalities. (2) PARA: This dataset provides diverse image and tabular attributes for personalized image aesthetics assessment. For the retrieval-based tasks, we employed the Shopee-Price Match Guarantee dataset (SPMG), which aims to determine if two products are identical, relying on data from image and text modalities. Our performance metrics include accuracy for multiclass classification tasks, the area under the ROC curve (AUC) for binary classification tasks and retrieval tasks, and the root mean square error (RMSE) for regression tasks. We also evaluated AutoM³L on a large number of unimodal datasets from the AutoML Benchmark[10] available from OpenML[2], which cover regression and binary/multiclass classification tasks. The metrics include logarithmic loss (Log-Loss) for multiclass classification tasks, the root mean square error (RMSE)

for regression tasks, and the area under the ROC curve (AUC) for binary classification tasks.

*3.1.2 Baseline.* Given the scarcity of specialized multimodal AutoML frameworks, our experimental evaluations were exclusively performed using the AutoKeras[3] and AutoGluon framework. AutoKeras is dedicated to neural architecture search (NAS) and hyperparameter optimization for a given dataset. Setting up training pipelines in AutoGluon required meticulous manual configurations. This involved specifying which models to train and conducting an extensive pre-exploration to determine the suitable parameters for hyperparameter optimization, including their respective search ranges. It's crucial to highlight that the automation and intelligence levels of AutoGluon remain challenging to quantify, and in this research, we innovatively measure them through the user study from the human perspective. See Appendix E for detailed experimental settings.

*3.1.3 IRB Approval for User Study.* The user study conducted in this research has received full approval from the Institutional Review Board (IRB). All methodologies, protocols, and procedures pertaining to human participants were carefully reviewed to ensure they align with ethical standards.

### 3.2 Quantitative Evaluation

We first carried out quantitative evaluations, drawing direct comparisons with AutoKeras and AutoGluon, focusing on the modality inference, automated feature engineering, and the automated hyperparameter optimization modules. For modality inference evaluation, apart from the modality inference component, all other aspects of the frameworks are kept consistent. For feature engineering and

---

[2]https://www.openml.org/

[3]https://github.com/keras-team/autokeras

**Table 4: Evaluation on the hyperparameter optimization. AutoM³L's self-recommended search space rivals, and in some cases surpasses, manually tuned search spaces. \* means the result of manual corrections in modality inference.**

| Method | PAP↑ | MMSD↑ | PPC↓ | PARA↓ | SPMG↑ | CH-SIMS↑ |
|---|---|---|---|---|---|---|
| AutoKeras | 0.385(0.012) | 0.925(0.007) | 23.21(0.285) | 0.744(0.021) | / | / |
| AutoGluon w/o HPO | 0.415(0.011) | 0.958(0.004) | 17.78(0.307) | 0.568(0.019) | 0.985(0.003) | 0.543(0.032)* |
| AutoGluon w/ HPO | 0.442(0.008) | 0.963(0.004) | 17.60(0.217) | 0.561(0.015) | 0.990(0.002) | 0.564(0.026)* |
| AutoM³L | 0.440(0.012) | 0.967(0.004) | 17.47(0.211) | 0.563(0.016) | 0.992(0.003) | 0.591(0.027) |

hyperparameter optimization, we aligned the modality inference from AutoKeras and AutoGluon with the results of AutoM³L to analyze their respective impacts on performance. To enhance the robustness of our results, we performed 10-fold cross-validation experiments on all datasets. The accuracy is reported as the mean value with its corresponding standard deviation. Afterwards, we evaluate the pipeline assembly module in terms of intelligence and usability through user study in the next section, due to its inherent difficulty in quantitative evaluation.

*3.2.1 Evaluation for Modality Inference.* Table 2 depicts the comparative performance analysis between AutoGluon's modality inference module and our LLM-based modality inference approach across various multimodal datasets. Since AutoKeras utilizes manually predefined data modality for each column, we excluded it from the comparisons in this experiment. Within AutoGluon, modality inference operates based on a set of manually defined rules. For instance, an attribute might be classified as a categorical modality if the count of its unique elements is below a certain threshold. Upon observing the results, it's evident that AutoM³L offers accuracy on par with AutoGluon for most datasets. This similarity in performance can be primarily attributed to the congruence in their modality inference outcomes. However, a notable divergence is observed with the CH-SIMS dataset. Due to the manually defined rules in AutoGluon being unable to infer video modality and misclassified the "text" attribute as "categorical", the assembly of the training pipeline was hindered, resulting in the failure of the training task. We manually corrected the misinference of the "text" attribute in AutoGluon, achieving the accuracy of 0.543(0.032). In comparison, AutoM³L demonstrated a significantly superior accuracy, with a notable 3.2% improvement. Such a result highlights the robustness of our LLM-based modality inference approach, which effectively infers modality details from column names and their associated data through in-context learning with only a few examples, making it significantly more efficient than cumbersome manually designed rules.

*3.2.2 Evaluation for Feature Engineering.* Table 3 illustrates the comparisons of data preprocessing modules using AutoGluon and AutoKeras with our LLM-based automated feature engineering module on multimodal datasets. Given the completeness of these datasets, we randomly masked portions of the tabular data and manually introduced noisy features from unrelated datasets to assess the effectiveness of automated feature engineering. For datasets without tabular modality, only noise features are introduced. Note that, AutoGluon lacks a dedicated feature engineering module for multimodal data, making this experiment a direct assessment of

our automated feature engineering. We observed that automated feature engineering, which implements feature filtering and data imputation, effectively mitigates the impact of noisy data. Across all test datasets, automated feature engineering showed improvements, while AutoGluon and AutoKeras suffered from performance degradation as they struggled to handle noisy data. Since retrieval tasks and video modality inputs are not supported, we did not test AutoKeras on relevant datasets.

*3.2.3 Evaluation for Hyperparameter Optimization.* We also conduct experiments to evaluate the automated hyperparameter optimization module within AutoM³L. Contrasting with AutoKeras and AutoGluon, which often require users to manually define the hyperparameter search space, AutoM³L simplifies this process.

From Table 4, it's evident that the integration of hyperparameter optimization during the training phase contributes positively to model performance. Impressively, AutoM³L matches AutoGluon's accuracy on most datasets and, due to its effective utilization of video information, it has realized a 2.7% improvement on the CH-SIMS dataset. However, the standout advantage of AutoM³L lies in its automation. While AutoGluon requires a manual setup, which can often be tedious, AutoM³L significantly reduces the need for human intervention, providing a more seamless and automated experience. Another finding is that AutoKeras achieves lower accuracy on all datasets. In our analysis, we attribute it to the network structures obtained within its limited network search space, which lacks pretraining on large-scale datasets. In contrast, our approach leverages the strength of pretrained models by linking with open-source communities such as HuggingFace and Timm. This integration allows us to access more powerful pretrained models, contributing to the improved performance demonstrated in our work.

*3.2.4 Uni-Modal Scenario Evaluation.* Given that most AutoML frameworks currently focus on single-modality AutoML, to demonstrate the scalability of AutoM³L, we also evaluated AutoM³L on a large-scale single-tubular modality AutoML Benchmark[10] from OpenML. We compared it with a plethora of popular single-modal AutoML frameworks[7, 9, 11, 20, 22, 30, 31] on large and representative datasets from AutoML Benchmark covering binary classification, multiclassification, and regression tasks. The metrics include logarithmic loss (LogLoss) for multiclass classification tasks, the root mean squared error (RMSE) for regression tasks, and the area under the ROC curve (AUC) for binary classification tasks. We reported the mean and standard deviation based on 10-fold cross-validation.

**Table 5: Evaluation on single-modal datasets, denoted as $mean(std)^{fails}$(part1). The red values represent the best results achieved in all comparison frameworks.**

| Task ID | Task Name | Task Type | Task Metric | AUTOGLUON | AUTO-SKLEARN | AUTO-SKLEARN 2 | FLAML | GAMA |
|---|---|---|---|---|---|---|---|---|
| 146818 | australi... | binary | accuracy | 0.940(0.020) | 0.932(0.019) | 0.940(0.020) | 0.939(0.025) | 0.940(0.019) |
| 146820 | wilt | binary | accuracy | 0.994(0.009) | 0.994(0.010) | 0.995(0.008) | 0.988(0.013) | 0.996(0.004) |
| 167120 | numerai2... | binary | accuracy | 0.524(0.005) | 0.530(0.005) | 0.531(0.004) | 0.528(0.005) | 0.532(0.004)[1] |
| 168757 | credit-g | binary | accuracy | 0.791(0.044) | 0.783(0.042) | 0.795(0.038) | 0.784(0.039) | 0.791(0.030) |
| 168868 | apsfailu... | binary | accuracy | 0.992(0.002) | 0.992(0.002) | 0.992(0.003) | 0.992(0.003) | 0.992(0.002) |
| 190137 | ozone-le... | binary | accuracy | 0.934(0.017) | 0.920(0.024) | 0.933(0.022) | 0.925(0.021) | 0.926(0.032) |
| 190411 | ada | binary | accuracy | 0.920(0.018) | 0.917(0.017) | 0.920(0.018) | 0.924(0.018) | 0.921(0.018) |
| 359955 | blood-tr... | binary | accuracy | 0.755(0.044) | 0.745(0.052) | 0.755(0.044) | 0.731(0.066) | 0.757(0.049) |
| 359956 | qsar-bio... | binary | accuracy | 0.941(0.035) | 0.929(0.036) | 0.937(0.027) | 0.928(0.033) | 0.937(0.032) |
| 359958 | pc4 | binary | accuracy | 0.951(0.018) | 0.941(0.020) | 0.949(0.017) | 0.949(0.019) | 0.951(0.019) |
| 359965 | kr-vs-kp | binary | accuracy | 1.000(0.000) | 1.000(0.000) | 1.000(0.000) | 1.000(0.000) | 1.000(0.000) |
| 359930 | quake | regression | rmse | 0.19(0.0093) | 0.19(0.0089) | - | 0.19(0.0091) | 0.19(0.0092) |
| 359931 | sensory | regression | rmse | 0.67(0.061) | 0.69(0.051) | - | 0.69(0.054) | 0.68(0.055) |
| 359933 | space ga | regression | rmse | 0.094(0.013) | 0.1(0.025) | - | 0.1(0.015) | 0.096(0.019) |
| 359939 | topo21 | regression | rmse | 0.028(0.0049) | 0.028(0.0049) | - | 0.028(0.0048) | 0.028(0.0048) |
| 359944 | abalone | regression | rmse | 2.1(0.12) | 2.1(0.11) | - | 2.1(0.12) | 2.1(0.1) |
| 359946 | pol | regression | rmse | 2.6(0.29) | 3.3(0.35) | - | 3.6(0.37) | 3.7(0.3) |
| 359936 | elevators | regression | rmse | 0.0018(5.2e − 05) | 0.0019(7.3e − 05) | - | 0.002(6.5e − 05) | 0.0019(6.5e − 05) |
| 359954 | eucalypt... | multiclass | logloss | 0.690(0.053) | 0.716(0.047) | 0.704(0.061) | 0.779(0.121) | 0.700(0.057) |
| 2073 | yeast | multiclass | logloss | 1.015(0.087) | 1.043(0.080) | 1.015(0.084) | 1.011(0.083) | 1.019(0.081)[5] |
| 359960 | car | multiclass | logloss | 0.004(0.011) | 0.004(0.008) | 0.002(0.004) | 0.003(0.005) | 0.012(0.008) |
| 359964 | dna | multiclass | logloss | 0.106(0.027) | 0.116(0.032) | 0.111(0.025) | 0.106(0.029) | 0.106(0.028) |
| 359984 | helena | multiclass | logloss | 2.470(0.016) | 2.526(0.018) | 2.485(0.031) | 2.564(0.019) | 2.731(nan)[9] |
| 359993 | okcupid-... | multiclass | logloss | 0.559(0.009) | 0.567(0.007) | 0.563(0.008) | 0.562(0.008) | 0.568(0.007) |

**Table 6: Evaluation on single-modal datasets, denoted as $mean(std)^{fails}$(part2). The red values represent the best results achieved in all comparison frameworks.**

| Task ID | Task Name | Task Type | Task Metric | H2O AUTOML | LIGHT AUTOML | MLJAR | TPOT | AUTOM3L |
|---|---|---|---|---|---|---|---|---|
| 146818 | australi... | binary | accuracy | 0.934(0.020) | 0.944(0.021) | 0.940(0.024) | 0.936(0.024) | 0.961(0.017) |
| 146820 | wilt | binary | accuracy | 0.993(0.009) | 0.994(0.007) | 0.994(0.003)[5] | 0.985(0.025) | 0.999(0.002) |
| 167120 | numerai2... | binary | accuracy | 0.531(0.004) | 0.531(0.005) | 0.530(0.004) | 0.527(0.006) | 0.534(0.007) |
| 168757 | credit-g | binary | accuracy | 0.782(0.043) | 0.788(0.035) | - | 0.787(0.034) | 0.825(0.032) |
| 168868 | apsfailu... | binary | accuracy | 0.992(0.002) | 0.994(nan)[9] | 0.993(0.002)[6] | 0.989(0.003)[1] | 0.990(0.013) |
| 190137 | ozone-le... | binary | accuracy | 0.930(0.016) | 0.930(0.016) | 0.911(0.019)[8] | 0.916(0.026) | 0.950(0.015) |
| 190411 | ada | binary | accuracy | 0.921(0.017) | 0.922(0.018) | 0.921(0.018) | 0.917(0.018) | 0.920(0.021) |
| 359955 | blood-tr... | binary | accuracy | 0.760(0.029) | 0.749(0.055) | - | 0.754(0.043) | 0.792(0.045) |
| 359956 | qsar-bio... | binary | accuracy | 0.937(0.037) | 0.933(0.033) | 0.926(nan)[9] | 0.933(0.031) | 0.949(0.024) |
| 359958 | pc4 | binary | accuracy | 0.945(0.022) | 0.950(0.016) | 0.951(0.017) | 0.943(0.023) | 0.960(0.017) |
| 359965 | kr-vs-kp | binary | accuracy | 1.000(0.000) | 1.000(0.000) | 1.000(0.000)[7] | 0.950(0.158) | 1.000(0.000) |
| 359930 | quake | regression | rmse | 0.19(0.0094) | 0.19(0.0099) | 0.19(0.0093) | 0.19(0.0096) | 0.18(0.0106) |
| 359931 | sensory | regression | rmse | 0.7(0.062) | 0.69(0.061) | 0.67(0.043) | 0.68(0.054) | 0.7(0.063) |
| 359933 | space ga | regression | rmse | 0.097(0.012) | 0.1(0.017) | 0.099(0.018) | 0.099(0.018) | 0.1(0.011) |
| 359939 | topo21 | regression | rmse | 0.028(0.0049) | 0.028(0.0049) | 0.028(0.0048) | 0.028(0.0048) | 0.028(0.0042) |
| 359944 | abalone | regression | rmse | 2.1(0.11) | 2.1(0.12) | 2.1(0.12) | 2.1(0.11) | 2.1(0.10) |
| 359946 | pol | regression | rmse | 3.4(0.28) | 3.9(0.33) | 2.2(0.23) | 3.7(0.38) | 2.2(0.16) |
| 359936 | elevators | regression | rmse | 0.002(0.00013) | 0.002(5.7e − 05) | 0.0019(5.8e − 05) | 0.0019(6.4e − 05) | 0.0018(6.4e − 05) |
| 359954 | eucalypt... | multiclass | logloss | 0.702(0.087) | 0.695(0.058) | 0.646(0.054) | 0.752(0.130) | 0.677(0.069) |
| 2073 | yeast | multiclass | logloss | 1.040(0.091) | 1.038(0.094)[5] | 1.004(0.085) | 1.029(0.083)[5] | 0.995(0.095) |
| 359960 | car | multiclass | logloss | 0.001(0.001) | 0.002(0.002) | 0.002(0.003) | 1.450(3.004) | 0.001(0.001) |
| 359964 | dna | multiclass | logloss | 0.109(0.030) | 0.109(0.026) | 0.109(0.025) | 0.112(0.025) | 0.098(0.026) |
| 359984 | helena | multiclass | logloss | 2.794(0.018) | 2.504(0.014) | 2.575(0.021)[1] | 2.922(0.039) | 2.54(0.020) |
| 359993 | okcupid-... | multiclass | logloss | 0.567(0.008) | 0.560(0.009) | 0.563(0.008) | 0.569(0.009) | 0.565(0.007) |

In the experiment, we employed the user instruction: "*the model with the best performance on tabular data*" to drive MS-LLM for model selection, and FT-Transformer was chosen and integrated with other components to form a training pipeline for subsequent training. The results in Table 5 and Table 6 demonstrate AutoM³L's strong performance even in single-modality settings. Notably, in most frameworks compared, model ensemble techniques are employed to produce final predictions. However, in this experiment, AutoM³L solely utilized a single model for evaluation and achieved competitive results, even outperforming others on most experimental tasks. We will showcase more comparative test results in the upcoming open-source projects.

## 3.3 User Study

*3.3.1 Hypothesis Formulation and Testing.* To assess AutoM³L's effectiveness, we conducted a user study focused on whether the LLM controller can enhance the degree of automation within the multimodal AutoML framework. We formulated null hypotheses:

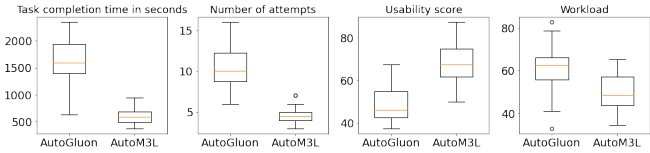

**Figure 5: Boxplots displaying the distribution of the four variables collected in the user study.**

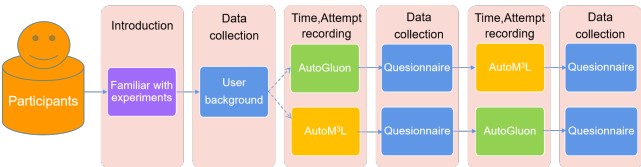

**Figure 6: The workflow of the user study to measure the user-friendliness of the AutoM³L.**

- **H1**: AutoM³L *does **not** reduce time required for learning and using the framework.*
- **H2**: AutoM³L *does **not** improve user action accuracy.*
- **H3**: AutoM³L *does **not** enhance overall framework usability.*
- **H4**: AutoM³L *does **not** decrease user workload.*

We performed single-sided t-tests to evaluate statistical significance. Specifically, we compared AutoM³L and AutoGluon on the following variables: task execution time, the number of attempts, system usability, and perceived workload.

*3.3.2 User Study Design.* As depicted in Fig. 6, our user study's workflow unfolds in structured phases. Note that the user study has been reviewed by IRB and granted full approval. The study begins with the orientation phase where voluntary participants are acquainted with the objectives, underlying motivations, and procedural details of the user study. This phase is followed by a user background survey, which gleans insights into participants' professional roles, their prior exposure to technologies such as LLM and AutoML, and other pertinent details. The core segment of the study involves hands-on tasks that participants undertake in two distinct conditions: perform multimodal task AutoML with AutoGluon and with AutoM³L. These tasks center around exploring the automation capabilities of the AutoML frameworks, as well as gauging the user-friendliness of their features such as hyperparameter optimization. Participants, guided by clear instructions, are tasked with constructing multimodal training pipelines employing certain models and defining specific hyperparameter optimization domains.

To ensure a balanced perspective, participants are randomly split into two groups: the first interacts with AutoGluon, while the second delves into AutoM³L. Upon task completion, the groups swap platforms. For a holistic understanding of user interactions, we meticulously track both the time taken by each participant for task execution and the number of attempts before the successful execution. The study culminates with a feedback session, where participants articulate their impressions regarding the usability and

**Table 7: Hypothesis testing results from paired two-sample one-sided t-tests.**

| Hypothesis | T Test Statistic | P-value | Null Hypothesis |
|:---:|:---:|:---:|:---:|
| H1 | 12.321 | $8.2 \times 10^{-11}$ | Reject |
| H2 | 10.655 | $9.3 \times 10^{-10}$ | Reject |
| H3 | -5.780 | $1.0 \times 10^{-5}$ | Reject |
| H4 | 3.949 | $4.3 \times 10^{-4}$ | Reject |

perceived workload of both AutoGluon and AutoM³L via questionnaire. Their feedback and responses to the questionnaire, captured using Google Forms, form a crucial dataset for the subsequent hypothesis testing and analysis. Our study cohort consisted of 20 diverse participants: 6 software developers, 10 AI researchers, and 4 students, which ensured a rich blend of perspectives of the involved users.

*3.3.3 Results and Analysis of Hypothesis Testing.* The data we gathered spanned four variables, visualized in Fig. 5. To validate our hypotheses, we performed paired two-sample t-tests (essentially one-sample, one-sided t-tests on differences) for the aforementioned variables across two experimental conditions: AutoGluon and AutoM³L. These tests were conducted at a significance level of 5%. The outcomes in Table 7 empower us to reject all the null hypotheses, underscoring the superior efficacy and user-friendliness of AutoM³L. The success of AutoM³L can be largely attributed to the interactive capabilities endowed by LLMs, which significantly reduce the learning curve and usage costs.

Since most researchers were familiar with LLMs but had limited AutoML experience, increasing their learning curve on AutoGluon. Whereas the majority of engineers and students were novices in both these spheres, facing steeper challenges in grasping AutoGluon. Interestingly, even researchers acquainted with AutoML felt that AutoM³L demonstrated superior ease of use comparatively. Collectively across backgrounds, AutoM³L attained higher user ratings, lower task completion times, and fewer failed attempts, which quantitatively validates its improved user-friendliness.

## 4 CONCLUSION

In this work, we introduce AutoM³L, an LLM-powered Automated Multimodal Machine Learning framework. AutoM³L explores automated pipeline construction, automated feature engineering, and automated hyperparameter optimization. This enables the realization of an end-to-end multimodal AutoML framework. Leveraging the exceptional capabilities of LLMs, AutoM³L provides adaptable and accessible solutions for multimodal data tasks. It offers automation, interactivity, and user customization. Through extensive experiments and user studies, we demonstrate AutoM³L's generality, effectiveness, and user-friendliness. This highlights its potential to transform multimodal AutoML. AutoM³L marks a significant advance, offering enhanced multimodal machine learning across domains. Our future direction is to encompass a diverse range of data modalities, spanning graph, audio, and point clouds, among others.

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
