# OpenReview forum: "AutoM3L: An Automated Multimodal Machine Learning Framework with Large Language Models"
_acmmm.org/ACMMM/2024/Conference — MM2024 Poster_

### Official Review · Reviewer_xU3V · 2024-05-01

**Rating:** 2
**Confidence:** 3

**Summary:**

The manuscript presents AutoM3L, a framework for automated multimodal machine learning. This framework leverages large language models (LLMs) to enhance the processes of modality inference, feature engineering, model selection, pipeline assembly, and hyperparameter tuning. Comparative analysis with frameworks such as AutoGluon suggests that the proposed method outperforms competitive baselines. Additionally, the study introduces a user study to evaluate the framework.

**Strengths:**

1. The paper explores the integration of LLMs within an AutoML framework, which is a significant contribution to the field.
2. The authors have conducted a user study to assess the usability of the proposed method, adding depth to the evaluation.
3. The framework demonstrates efficacy advantages on single-modal tasks, showcasing its versatility.

**Limitations:**

1. The paper does not clearly articulate the specific problem it aims to address, instead listing a series of optimizations targeting AutoGluon, which are trivial.
2. There is a lack of ablation studies to evaluate the impact of individual modules.
3. The improvements in performance on multimodal data compared to AutoGluon are marginal.
4. The linked code repository contains empty code folders, which is not conducive to reproducibility. The authors should populate the repository with the actual source code.
5. Given the significant financial cost associated with LLMs, there is a need for a discussion on cost, which is currently absent.
6. If the primary contributions are attributed to the LLMs, a comparison of different LLMs should be discussed in terms of their effectiveness.
7. The quality of figures is suboptimal; for example, the figure in the upper left corner of Figure 1, and Figure 4(a) are not clearly rendered.
8. There are typographical errors in Figure 1; "Genger" should be corrected to "Gender", and "Hyperparamter" should be corrected to "Hyperparameter".
9. Figure 3 requires additional specifics for coherence: What is the precision of the mentioned retrieval? How many models are included in the model zoo, and what is the distribution of model usage within the zoo?

**Suitability:**

2

---

### Official Review · Reviewer_wRVH · 2024-05-02

**Rating:** 4
**Confidence:** 3

**Summary:**

The paper proposes an automated multimodal machine learning framework by using large language models as the procedure controller, including dataset organization, modality inference module, feature engineering module, model selection module, pipeline assembly module, and hyperparameter optimization module. Experiments on various multimodal and single-modal datasets are conducted to show the effectiveness of the framework. Further user study demonstrates the automation of the framework.

**Strengths:**

1. The paper proposes an automated multimodal machine learning framework by using large language models as the procedure controller.
2. The paper is well-written and the experiments are comprehensive.

**Limitations:**

1. The method lacks some novelty because some previous works [1][2] have done something similar to this paper.

    [1] HuggingGPT: Solving AI Tasks with ChatGPT and its Friends in Hugging Face

    [2] Visual ChatGPT: Talking, Drawing and Editing with Visual Foundation Models

2. The evaluation metrics in this paper are all end-to-end. Are there any modular evaluation metrics or ablation experiments to demonstrate the effectiveness of each design module?

3. The efficiency issue of large language models. Is there any advantage in efficiency of this paper compared to traditional methods, as invoking large language models requires high costs and time?

**Suitability:**

3

---

### Official Review · Reviewer_tf5a · 2024-05-29

**Rating:** 6
**Confidence:** 3

**Summary:**

This paper introduces an automated multimodal machine learning framework called AutoM3L, which leverages large language models (LLMs) as controllers to automatically construct multimodal training pipelines. The authors point out that AutoML reduces the need for manual intervention in the machine learning process, but most AutoML solutions primarily focus on unimodal data. AutoM3L automates five functional components of the machine learning pipeline using LLMs: modality inference, automated feature engineering, model selection, pipeline assembly, and hyperparameter optimization. The authors conducted experiments on six multimodal datasets, including classification, regression, and retrieval tasks. Compared with the AutoKeras and AutoGluon frameworks, AutoM3L demonstrated competitive or superior performance on most datasets. A user study was conducted to evaluate the usability and effectiveness of AutoM3L. The results of the user study supported AutoM3L's advantages in reducing the learning curve and enhancing user-friendliness.

**Strengths:**

Automated multimodal machine learning (AutoM3L) is indeed an important research area. Automating this process can significantly reduce the workload of data scientists and machine learning engineers in model selection, feature engineering, and hyperparameter tuning. As far as I know, there is currently no widely recognized open-source platform specifically designed for automating multimodal machine learning (although there are some tools and libraries available for automating multimodal machine learning tasks, they may not be specifically designed for multimodal data).

The authors conducted experiments on six multimodal datasets, including classification, regression, and retrieval tasks. Compared with the well-known frameworks AutoKeras and AutoGluon, AutoM3L demonstrated competitive or superior performance on most datasets.

**Limitations:**

The fusion of multimodal data is a crucial step in multimodal data processing. In this paper, the AutoM3L framework employs a component called Pipeline Assembly-LLM (PA-LLM) which generates executable code for model fusion and data processing based on the provided model configuration files. However, the authors did not specifically validate the performance of this module in their experiments. Providing dedicated case studies and performance evaluations of this module would be beneficial in understanding its effectiveness and potential impact on the overall framework's performance.

**Suitability:**

3

---

### Official Review · Reviewer_FaqR · 2024-06-06

**Rating:** 4
**Confidence:** 2

**Summary:**

- This paper introduces AutoM3L, a framework that automates the development of machine learning pipelines for multimodal data by leveraging LLMs.
- The framework enhances the process of feature engineering, model selection, and hyperparameter optimization, reducing the need for manual intervention.
- Comprehensive experiments on diverse multimodal and unimodal datasets demonstrate that AutoM3L achieves competitive or superior performance compared to conventional rule-based AutoML frameworks like AutoGluon.
- Additionally, a user study highlights the user-friendliness and reduced learning curve of AutoM3L compared to rule-based AutoML methods, enabled by the interactive capabilities of LLMs.

**Strengths:**

- The paper introduces a novel approach to multimodal AutoML by harnessing the capabilities of large language models (LLMs) as controllers for automating various stages of the machine learning pipeline.
- The paper demonstrates the effectiveness of AutoM3L through extensive experiments on diverse multimodal and unimodal datasets.
- The paper is well-structured and clearly written, providing a detailed description of each component of the framework and its role within the overall system.

**Limitations:**

- The paper does not fully address the computational costs associated with using LLMs, especially when considering large-scale applications or real-time processing needs.
- While AutoM3L automates the machine learning pipeline construction process, the paper does not extensively discuss the interpretability and explainability of the generated pipelines. Providing insights into how LLMs make decisions during various stages, such as model selection and hyperparameter optimization, would enhance the transparency and trustworthiness of the framework.
- The user study, though insightful, appears limited in scope with only 20 participants. A larger and more diverse participant pool might provide more robust evidence of usability and learning curve improvements.
- Moreover, the metrics used to evaluate user interaction could be expanded beyond time and accuracy to include qualitative aspects of user satisfaction and cognitive load, which would provide a deeper understanding of the user experience.

**Suitability:**

2

---

### Meta-Review · Program_Chairs · 2024-06-27

**Recommendation:** Accept (Poster)
**Confidence:** 4

**Metareview:**

This paper introduces AutoM3L, an Automated Multimodal Machine Learning framework leveraging LLMs as controllers to automatically construct multimodal training pipelines, which simplifies user engagement and shows competitive or superior performance on diverse datasets and in user-friendliness compared to traditional methods. However, the reviewers raised concerns but not fully addressed in the rebuttal including:
1) This work is based on FT-Transformer in AutoGluon. It enabled the hyperparameter optimization of FT-Transformer and made some trivial optimizations.  This has not made a significant contribution to any of the fields of AutoML.
2) The experimental scope is restricted to a limited range of datasets with relatively simple features, and the study lacks ablation tests to evaluate the impact of various components.
3) The performance improvement in multimodal tasks is not significant compared to AutoGluon. The experimental comparison is confined to a single baseline, AutoGluon, and does not adequately address or contrast with other relevant works in multimodal AutoML or Neural Architecture Search (NAS).

The AC has throughly checked the paper and the reviews, and agree to these concerns. In conclusion, although the paper covers an interesting topic, the lack of depth in research insights and comprehensive evaluation reduces its overall contribution to the field. Considering the intense competition at ACM MM, the AC suggests that the authors revise the paper based on the reviewers' feedback and consider submitting it to the next conference with best wishes.

*** TPC Meta-Review Addendum ***
This paper is very much on the borderline for acceptance. Besides the official reviews, there was significant discussion about this paper between the AC, SAC, and the TPC. The paper has limitations in terms of its enhancements over AutuGluon and the limited performance gains in different tasks. At the same time, multiple reviewers have recognized its strengths as an end to end framework, and its value as a tool that enables multimedia research by a broader set of researchers. The TPC is suggesting that the paper be accepted as a poster to allow for more conversation on this topic. The tool, if useful for a small percentage of others, will already be a positive step forward.